# Effects of Primary Processing and Pre-Salting of Baltic Herring on Contribution of Digestive Proteases in Marinating Process

Patryk Kamiński [1,*], Mariusz Szymczak [1] and Barbara Szymczak [2]

1   Department of Toxicology, Dairy Technology and Food Storage, Faculty of Food Sciences and Fisheries, West Pomeranian University of Technology in Szczecin, Papieza Pawla VI 3, 71-459 Szczecin, Poland
2   Department of Microbiology, Faculty of Food Sciences and Fisheries, West Pomeranian University of Technology in Szczecin, Papieza Pawla VI 3, 71-459 Szczecin, Poland
*   Correspondence: patryk-kaminski@zut.edu.pl; Tel.: +48-91-449-65-01

**Abstract:** The low-technological quality of herring caught during the feeding season makes it impossible to achieve full ripeness of the meat in marinades. One solution may be to assist ripening using herring digestive tract proteases. Therefore, whole herring, headed herring and fillets were marinated for 2–14 days using the German (direct) and Danish (pre-salted) methods. The results showed that the mass of marinades from fillets was lower than from herring with intestines and correlated strongly with salt concentration in the Danish method, in contrast to the German method. Marinades from whole and headed herring had significantly higher trypsin, chymotrypsin, carboxypeptidase-A and cathepsin activities than marinated fillets. The herring marinated with viscera had 2–3 times higher non-protein nitrogen, peptide and amino acid fractions, as well as ripened 3 days faster than the marinated fillets. After 2 weeks of marinating, the fillets did not achieve full ripeness of the meat, unlike marinades made from whole and headed herring. The pre-salting stage in the Danish method significantly reduced cathepsin D activity by the tenth day of marinating, which was compensated by digestive proteases only in the case of whole or headed herring. The digestive proteases activity in the fillets was too low to achieve the same effect. Sensory evaluation of texture and hardness-TPA correlated strongly with several proteases in whole herring marinades, in contrast to a weak correlation with only one protease when marinating fillets. Marinating with intestines makes it possible to produce marinades faster, more efficiently and with higher sensory quality from herring of low-technological quality.

**Keywords:** herring; marinades; primary processing; pre-salting; cathepsins; intestines proteases

## 1. Introduction

The high degree of ripeness of the meat of herring marinades promotes high sensory quality and nutritional value, and facilitates digestion, especially in the elderly or those with certain gastrointestinal diseases. Cold-ripened marinades are produced mainly from herring, the technological quality of which varies seasonally [1]. Due to the high price of the raw material, the industry tries to marinate cheaper herring caught at the beginning of the feeding season, resulting in low ripeness of marinated meat. The ripening of marinades is mainly related to cathepsins D, L and B activity [2,3]. Herring caught during heavy feeding season have low cathepsins activity, and therefore, are worse suited for marinades production. Higher cathepsins activity causes better ripeness of meat and the formation of a higher content of peptides and free amino acids, which provides sensory and nutritional characteristics to marinades. Protein hydrolysis products (PHP) released from marinated herring proteins also show biological activity [4,5]. These peptides may have antihypertensive, anticoagulant, antioxidant or even immunomodulatory effects [6].

Kamiński et al. [7] showed that cold storage of herring prior to marinating allows the use of digestive proteases to improve the low technological quality of the raw material.

During the 7 days in cold storage, trypsin, chymotrypsin and carboxypeptidases diffused from the viscera into the herring muscle, which started proteolysis of the meat and increased cathepsin activity during marinating. The result was an increase in the sensory evaluation of marinades for taste, aroma and texture. The disadvantages of this method are the requirement of additional time before marinating and a lower yield of marinade mass. Currently, Baltic herring after primary processing (heading, filleting) are marinated in two ways. The longest known method, called German or classic or direct method, consists of marinating of fillets in acid-salty brine for 7 days. This method has a 75–85% mass yield of the marinating process [8]. The second method, called the Danish method, has a 90–100% mass yield and is based on wet salting of fillets for 2 days, followed by marinating in an acid-salty brine [9]. Despite the fact that the Danish method has been known in the industry for many years, there is a lack of research regarding the effect of pre-salting on protease activity in the process of marinating herring.

Currently, cold marinades are mainly made from herring fillets, but marinated sprats in northern and central Europe and marinated sardines in southern Europe and western Asia are also available on the market. Baltic herring caught at the beginning of the feeding season are characterized by a small size, similar to that of a large sprat [10]. Large sprats are subjected to nobbing before marinating, which involves separating the head along with the digestive tract, without opening the abdominal cavity and without removing the pyloric caceae and gonads [11]. The pyloric caceae contain very active proteases [7,12]. Practice shows that mechanical nobbing is not always effective and some of the sprat is marinated with whole intestines. There are no results in the literature for the marinating method of whole or nobbed herring, especially in combination with the Danish marinating method, in which pre-salting may change the activation of digestive proteases, as occurs during traditional salting of herring [13,14].

Protease activity is crucial in the ripening process of marinated herring meat, but so far, protease activity has only been studied during the classic marination of fillets [2,15]. The effect of pre-salting on protease activity in marinades is unknown, while the contribution of digestive proteases during marination has been described in only one publication [7]. Therefore, the aim of this study is to determine the effect of primary processing of Baltic herring (whole, headed, fillets) with and without pre-salting stage on contribution of digestive and cathepsins proteases during ripening of marinades using Danish and German methods.

## 2. Materials and Methods

### 2.1. Baltic Herring and Marinating Methods

Baltic herring (*Clupea harengus membras*) were caught in December (FAO 27IIId), transported within 24 h in a polystyrene ice box to the laboratory, where Quality Index Method (QIM) and morphometry of whole herring were assessed [7,16]. The fish condition and fullness of the herring's stomachs indicated foraging. The gonads had a mass of $11.3 \pm 4.0$ g and maturity stage III on the Maier scale, while the weight of the digestive tract was $6.5 \pm 1.8$ g. The herring were $22.3 \pm 0.8$ cm long and weighed $113.0 \pm 28.2$ g, with meat containing $11.7 \pm 0.1\%$ lipids, $71.9 \pm 0.1\%$ water and $15.9 \pm 0.3\%$ protein.

The herring were divided into three raw material groups: whole herring, headed herring and fillets (control sample), which were marinated by the (G-) German or (D-) Danish method obtaining marinated samples: G-whole, G-headed, G-fillets, and D-whole, D-headed, D-fillets (Figure 1). The German method was performed by marinating each group of herring separately in 5% vinegar and 6% rock salt solution for 14 days [2]. The Danish method was performed using salting for 2 days in 10% brine, followed by marinating in 5% vinegar and 1.8% rock salt solution for 12 days [9]. All herring samples were marinated at 7 °C; fish to brine ratio was 1:1 (*w:w*). After 2, 4, 7, 10 and 14 days of marinating, the fish were transferred to a sieve and herrings were weighed.

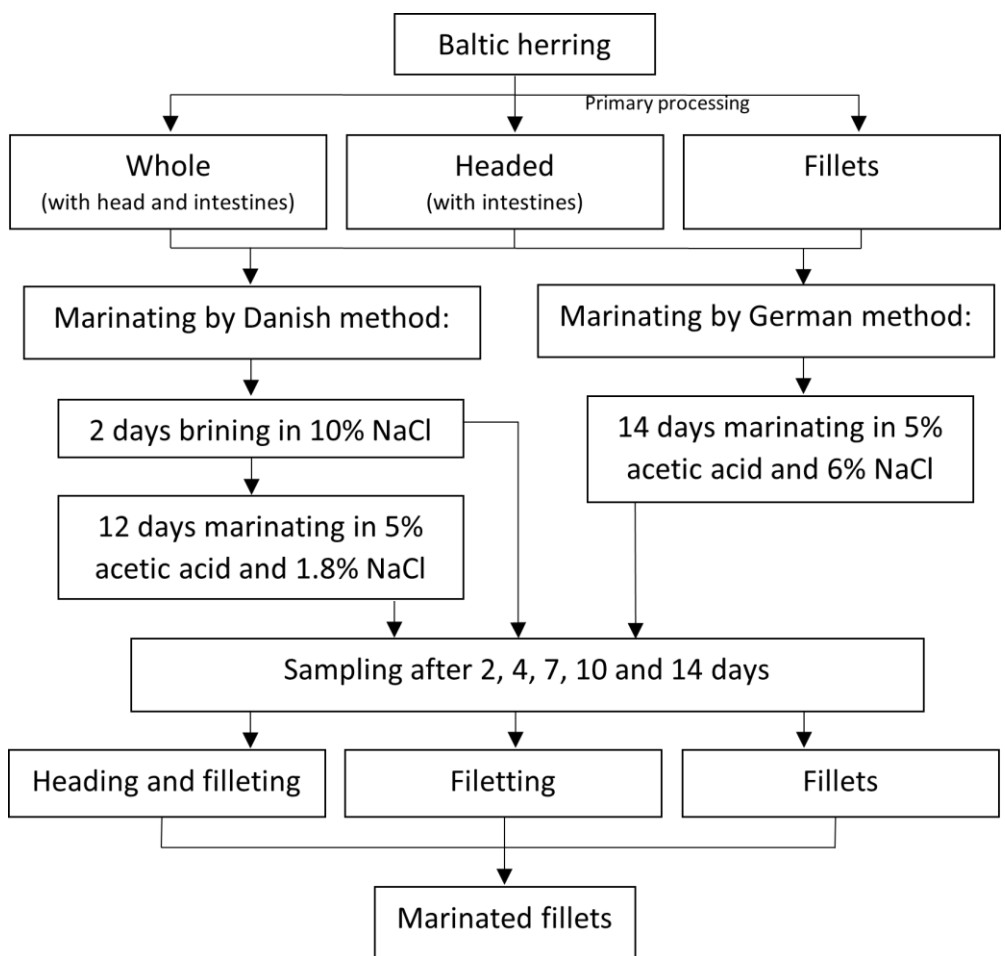

**Figure 1.** Diagram showing the performance of marinades using the German and Danish methods.

### 2.2. Total Nitrogen, Lipids, Salt, pH, Acidity and Meat Moisture Analysis

The marinated herring fillets were skinned and minced. The pH value in the meat with distilled water homogenate (1:10, *w:v*) was determined using a digital pH-meter. Total nitrogen Kjeldahl's method, total lipids Soxhlet method, moisture, salt contents and total acidity were determined using standard AOAC analytical techniques no. 940.25, 960.39, 950.46B and 976.09, respectively.

### 2.3. Extractions and Activity Assay of Proteases

Digestive protease extracts were made from fresh (raw material) and marinated herring meat using water (1:10, *v:w*) and TrisHCl 150 mM buffer at pH 7.8 (to buffer acetic acid in marinades), respectively [7]. Cathepsins from raw material and marinated meat were extracted with ultra-pure water. Samples were homogenized for 30 s at 34,000 rpm and centrifuged after 30 min (10 min, 9000× *g*, 4 °C). Obtained supernatant was centrifuged at 20,000× *g* (10 min, 4 °C) to obtain the crude protease extract, where digestive proteases and cathepsins activity were determined. Amidase trypsin (Am-Tr) was measured at pH 8.2 against N-benzoyl-DL-arginine (BApNA) [17]; Esterase trypsin (Es-Tr) at pH 8.2 against Nα-*p*-Tosyl-L arginine methyl ester hydrochloride (TAME) and Chymotrypsin at pH 7.8 against N-Benzoyl-L-tyrosine ethyl ester (BTEE) [18]; Carboxypeptidase-A (Cp-A) at pH 7.5 against 4-Methoxyphenylazoformyl-Phe (AAFP) [19]. The activity of cathepsin B against Z-RR-MCA, cathepsin D against Mca-GKPILFFRLK(Dnp)-r-NH2 with and without pepstatin-A, and cathepsin L against Z-FR-MCA were determined according to [2]. Protease activity assays were performed at 37 °C using continuous spectrophotometric rate determination,

and units of enzyme activity were converted per 1 g of tissue. Chemicals from PeptaNova (Concord, CA, USA) and ultrapure water were used for analysis.

### 2.4. Non-Protein Nitrogen Fraction Analysis

Determinations of (a) non-protein nitrogen by Kjeldahl's method (AOAC no. 940.25), (b) peptides [PHP(R)] and amino acids [PHP(A)] fractions by Lowry's methods as modified by [20], were carried out in 50 g·kg$^{-1}$ trichloroacetic acid (TCA) extract.

### 2.5. Texture Profile Analysis (TPA)

Hardness-TPA was determined in four fillets from each sample with a TA-XTplus Texture Analyzer (Stable Micro Systems, Godalming, UK). TPA tests consisted of twice compressing samples in the same spot with a cylindrical probe P10 (10 mm) up to 50% of fillet height at the speed of 5 mm·s$^{-1}$, and were conducted in three replications for each fillet separately (12 replications in each sample), and only on the dorsal muscle in an area from 2/10 to 6/10 fillets length measured from the head side. Their courses were recorded as curves representing changes of force in time and expressed in newtons, N.

### 2.6. Sensory Profiling

Marinated meat was analyzed by sensory profiling performed by a trained sensory panel, using a five-point unstructured scale with a 0.5-point accuracy anchored at their extremes with minimum and maximum degrees of acceptance [21]. A higher note signified better sensory attributes (0 points being the worst sensory/extremely disliked; 5 points being the best sensory/extremely liked). Four skinned fillets from each sample were served in porcelain trays. The assessors used water and flat bread to clean their palates between samples. Sensory attributes were: texture, flavor, odor and appearance. The sum of individual scores gave a total score that represented the overall sensory evaluation of the marinated fillets. The seven panelists participating in the sensory evaluation of marinades in this study had many years of experience in the subject of marinating fish. In addition, before starting the research, the panelists were trained on the sensory assessment methods described above.

### 2.7. Microbiological Analysis

Three fillets from marinated herring were taken for analysis using disinfected (peracetic acid, 30% $H_2O_2$) knives and plastic boards. The fillets were held with sterile tweezers and rinsed for 1–2 s with cold tap water to remove any remaining viscera and blood. The preparation of the test samples was performed according to the ISO standard [22]. Samples in triplicate were weighed at 20 g each into sterile stomacher bags and 180 mL of sterile dilution fluid (P-0061, BTL, Warsaw, Poland) was added. Samples were homogenized (BagMixer 400P, Interscience, Saint Nom la Bretêche, France), and then dilutions and surface cultures were performed on nutrient agar medium (P-0075, BTL, Poland) for total psychrophilic bacteria [23], and on Sabouraud medium for total yeast and mold [24]. In addition, flooded cultures were performed on VRBGLA medium (BT5158.02, Biomaxima, Lublin, Poland) for *Enterobacteriacea* [25] and on nutrient agar (01140, Scharlab, Barcelona, Spain) for total mesophilic bacteria [26].

### 2.8. Statistical Analysis

Results were statistically analyzed using one-way analysis of variance (ANOVA) with Statistica 13.3 software (Statsoft, Tulsa, OK, USA). The differences between treatments were examined using analysis of variance (ANOVA) and the post hoc Tukey's test of honestly significant differences ($p < 0.05$) [27]. Correlations between marinating time, proteases activities, proteolysis indicators and sensory analyses for each marinating method and raw materials were separately investigated using Principal Component Analyses (PCA) [28,29]. All analyses were performed in triplicate (n = 3), except TPA (n = 12), and results were presented as mean and standard deviation.

## 3. Results and Discussion

### 3.1. Mass Yield, pH and Salt Concentration

The mass of herring marinated by the German method (G-) decreased by day 14 (Figure 2A), which was caused by a decrease in meat pH close to the isoelectric point and in the resulting loss of water (Figure 2B). The mass of German marinades significantly correlated with marinating time, most strongly for G-whole r = −0.987 (Figure 3A). In the case of the Danish (D-) method, the yield increased after 2 days due to protein salting and moisture increase, but when acetic acid was added, fish mass and moisture significantly decreased (Figure 2A,B). The mass of Danish marinates also significantly correlated with marinating time, but the correlation was lower than that of German marinates. After 4–7 days of marinating, the mass yield of G-fillets marinades averaged 78.7%, while G-headed and G-whole marinades had higher yields of 4.1 and 15.0 percentage points, respectively (Figure 2A). At the same time, D-fillets marinades had an average mass yield of 83.3%, while D-headed and D-whole marinades were 4.7 and 9.1 percentage points higher, respectively. Thus, the advantage of higher mass yield of the Danish method over the German method when marinating fillets [9] was not present in the case of marinating whole herring.

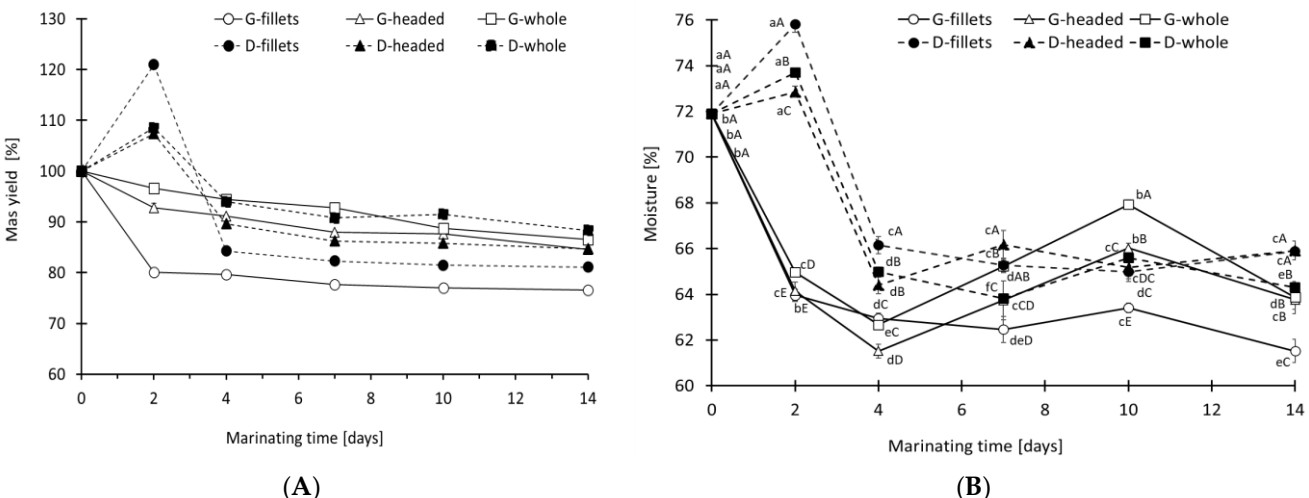

**Figure 2.** Effect of marinating method and degree of primary processing of Baltic herring (whole, headed, fillets) on (**A**) mass yield and (**B**) moisture during marinating. G—German method, D-—Danish method. [a,b,c] Results marked with the same lower case letter do not statistically differ by the influence of marinating time; [A,B,C] Results marked with the same capital letter do not statistically differ by the influence of the marinating method.

G-fillets marinades after 4–7 days of marinating had an average meat pH value of 3.91, while G-headed and G-whole had 4.16 and 4.26, respectively (Figure 4A). The pH value significantly correlated with the decreasing mass of German marinades: G-whole (0.973), G-fillets (0.891) and G-headed (0.871) (Figure 3A,C,E). The D-fillets marinades had a meat pH of 3.88, while the D-headed and D-whole marinades had a pH of 4.22 and 4.38, respectively. The pH values of D-fillets, D-headed and D-whole marinades significantly correlated with a mass yield of 0.991, 0.877 and 0.750, respectively (Figure 3B,D,F). The content of table salt after 4–7 days of marinating in the meat of G-fillets marinades was 2.96% and was 0.07–0.17% higher than in G-whole and G-headed marinades (Figure 4B). For marinating using the Danish method, the differences in salt content between D-fillets marinades (2.80%) and D-headed (2.38%) and D-whole (2.16%) marinades were higher than that of the German method (Figure 4B). PCA analysis showed that the higher the degree of primary processing of the herring (whole < headed < fillets), salt concentration correlated less strongly with the mass yield of marinades performed by the German method (G-whole −0.820, G-headed −0.801, G-fillets −0.469) (Figure 3A,C,E), in contrast to Danish

method (D-whole 0.750, D-headed 0.877 and D-fillets 0.991) (Figure 3B,D,F). Therefore, in marinades made of whole herring, the skin reduced the loss of water and protein from the meat [30], and also reduced the diffusion of acetic acid and salt into the meat (Figure 4A,B), which synergistically reduced the water holding capacity of the marinated meat [31].

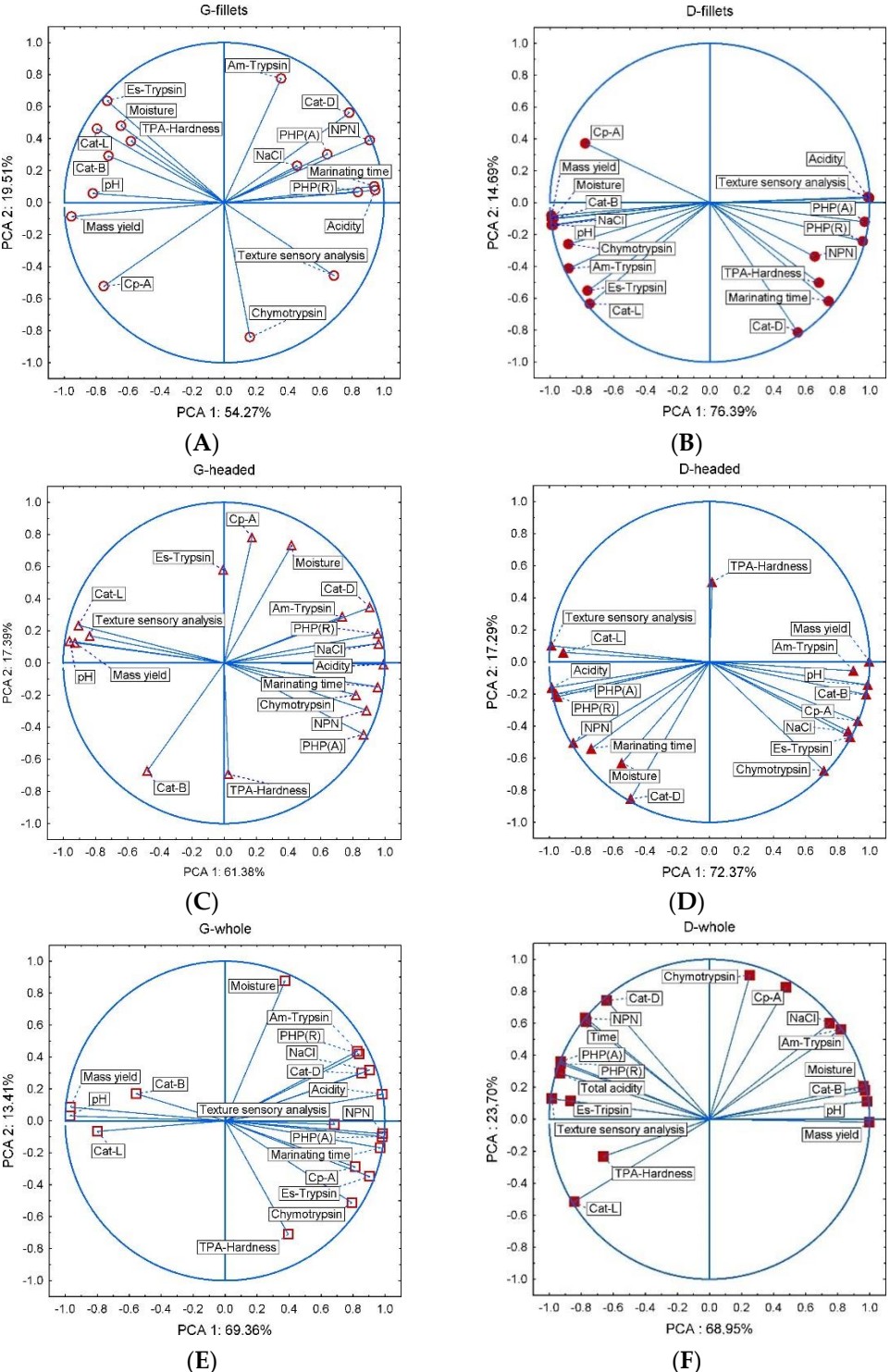

**Figure 3.** PCA biplot for correlations between protease activity, physicochemical parameters and sensory analyses of marinades depending on marinating methods: (**A,C,E**) German method (G-), (**B,D,F**) Danish method (D-) and depending on degree of primary processing of Baltic herring: (**A,B**), fillets of herring, (**C,D**) headed herring and (**E,F**) whole herring.

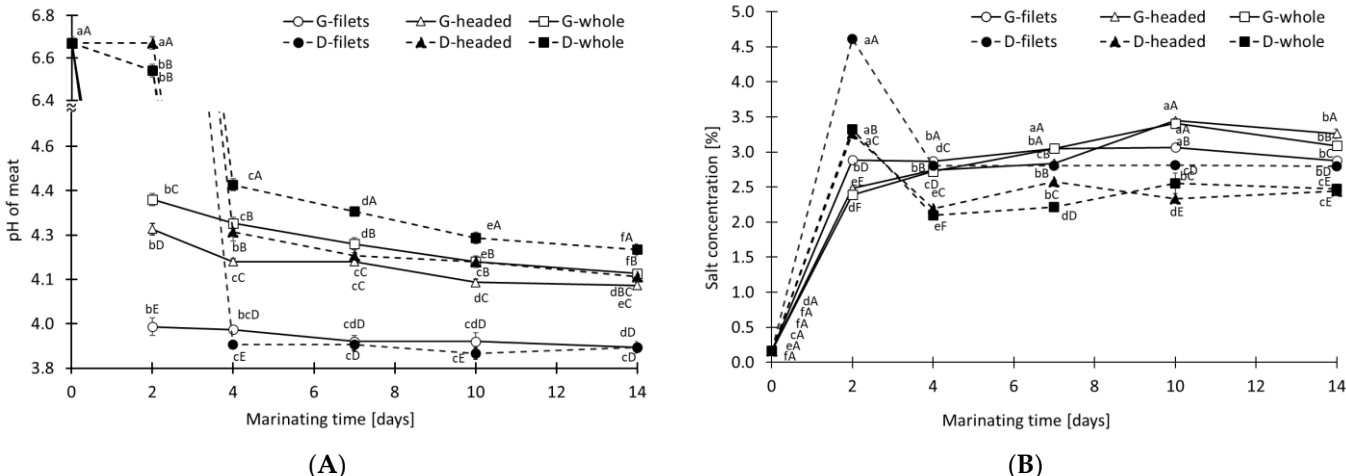

**Figure 4.** Effect of marinating method (G—-German, D—-Denmark) and degree of primary processing of Baltic herring (whole, headed, fillets) on (**A**) mass yield and (**B**) moisture during marinating. [a,b,c] Results marked with the same lower case letter do not statistically differ by the influence of marinating time; [A,B,C] Results marked with the same capital letter do not statistically differ by the influence of the marinating method. For a better presentation of the results, a break ($\approx$) in the Y axis was used.

### 3.2. Activity of Digestive Proteases and Cathepsins

During marinating, mainly muscle proteases—cathepsins—were active, while during marinating whole and headed herring, digestive proteases such as trypsin, chymotrypsin and carboxypeptidases were also diffused into the meat. The results showed that digestive proteases were already active in fresh herring meat (0 day of marinating) and their activity decreased during marinating (Figure 5), mainly due to a decrease in pH [7]. After the fourth and tenth days of marinating, trypsin esterase activity was higher in the marinades made using the German method than the Danish method, and G-whole marinades had the highest activity after 4–14 days of marinating (11.95–20.12 U) (Figure 5A). Chymotrypsin activity was lower in marinades from fillets than from whole or headed herring, with the exception of day 4 (Figure 5C). Chymotrypsin activity decreased the most after 2 days of marinating using the German method, or after 4 days of marinating using the Danish method. On subsequent days, chymotrypsin activity increased, especially in marinades made from whole and headed herring, and more so in Danish than German marinades. Additionally, carboxypeptidase-A activity decreased by 2–4 days of marinating, except for fillets, where activity decreased by 10–14 days (Figure 5D). Between 4 and 14 days of marinating, an increase in carboxypeptidase-A activity was noted in marinades made using both methods from whole herring. D-whole and G-whole marinades from days 7 to 14 had the significantly highest carboxypeptidase-A activity, 9.22–9.54 U and 6.89–8.95 U, respectively. Marinating time significantly and positively correlated with trypsin esterase activity in G-whole (0.956) and D-whole (0.726) (Figure 3E,F). There was also a significant positive correlation between marinating time and all digestive proteases in G-whole marinades (Figure 3E), except for Cp-A activity, which negatively correlated with marinating time in fillets (G-fillets −0.842, D-fillets −0.739) (Figure 3E,F). Salt concentration and pH values in marinades significantly correlated with all proteases in G-whole and D-whole marinades (Figure 3E,F), and in D-fillets and D-headed marinades (Figure 3B,D). It can be noted that the marinating of whole herring with intestines contributed to greater digestive protease activity in meat than the marinating of fillets obtained by cold-stored whole herring [7].

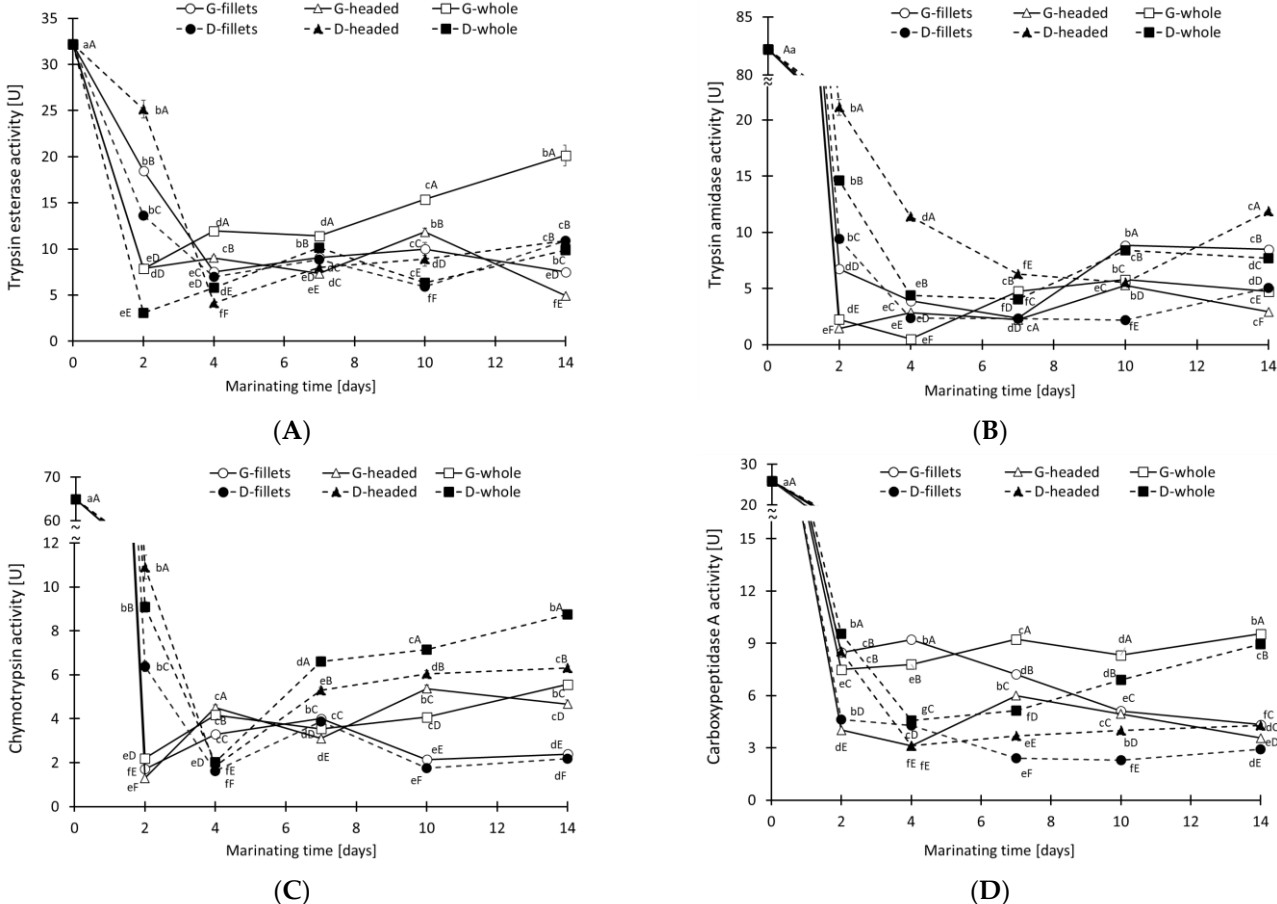

**Figure 5.** Effect of marinating method (G—-German, D—-Denmark) and degree of primary processing of Baltic herring (whole, headed, fillets) on activity of (**A**) trypsin esterase, (**B**) trypsin amidase, (**C**) chymotrypsin and (**D**) carboxypeptidase-A during marinating. [a,b,c] Results marked with the same lower case letter do not statistically differ by the influence of marinating time; [A,B,C] Results marked with the same capital letter do not statistically differ by the influence of the marinating method. For a better presentation of the results, a break ($\approx$) in the Y axis was used.

Cathepsin D activity in marinades made by the German method increased by 3–4 times after 4–7 days of marinating; while in marinades made by the Danish method, it increased by 2 to 3 times (Figure 6A). Therefore, cathepsin D activity significantly and positively correlated with marinating times in all samples (Figure 3). German marinades had maximum cathepsin D activity (6987–8051 U) after 10 days, while in Danish marinades, its activity increased until the fourteenth day, reaching 6913–7867 U. Low pH, along with the lowest possible salt concentration, are required to activate cathepsin D [2]. In G-headed and G-whole samples, cathepsin D activity increased with decreasing pH, as confirmed by a significant correlation, −0.793 and −0.861, respectively. In Danish marinades, after 2 days, the salt concentration was about 1.5 times higher than in German marinades, while the pH was close to neutral compared to strongly acidic in German marinades. Therefore, it is likely that the lowest cathepsin D activity in Danish marinades was due to the inhibitory effect of salt and the lack of activating acetic acid during the first two days of brining [32]. Cathepsin D activity is important for the ripening of marinated meat, because this endoprotease prepares substrates for the other cathepsins [33].

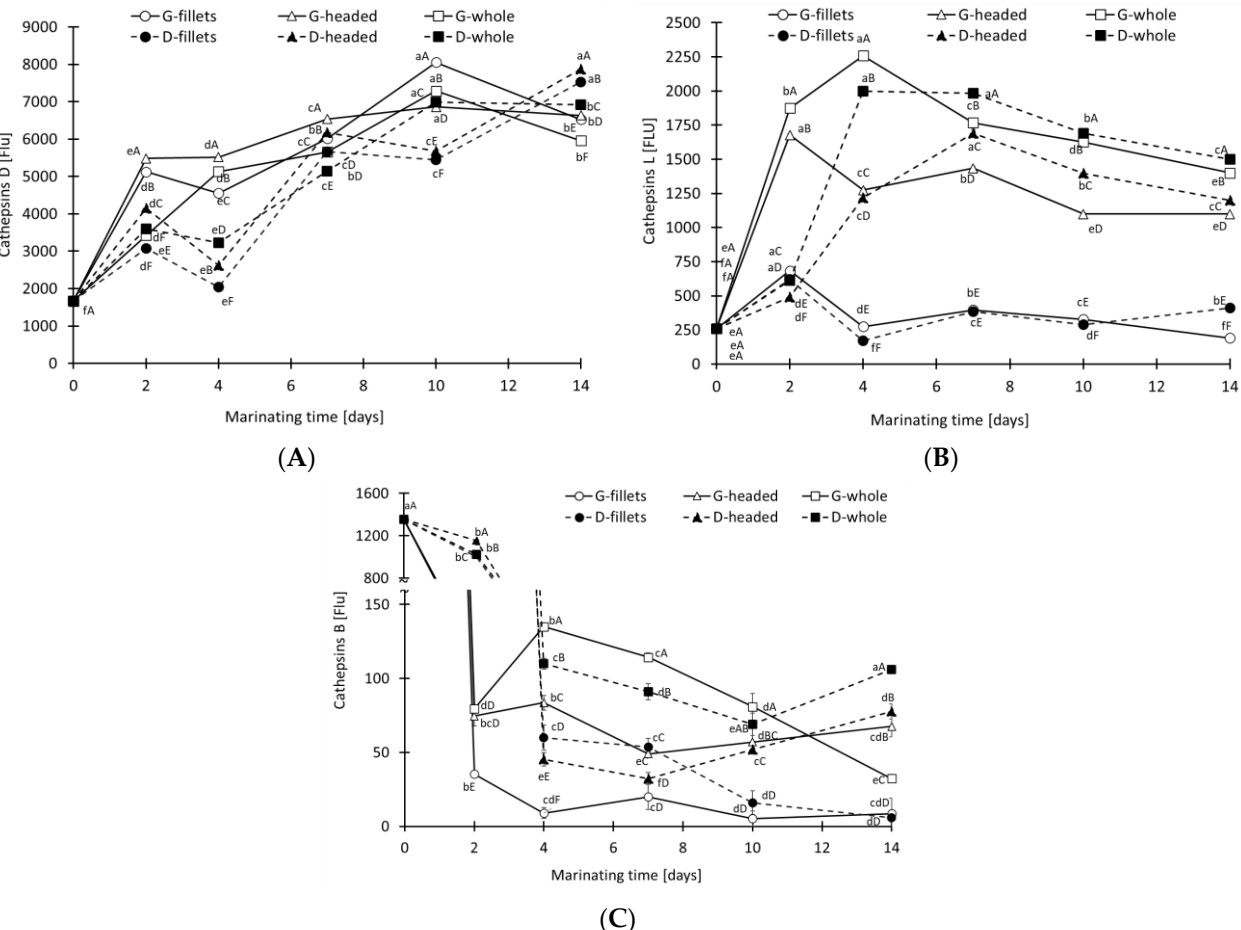

**Figure 6.** Effect of marinating method (G—-German, D—-Denmark) and degree of primary processing of Baltic herring (whole, headed, fillet) on activity of (**A**) cathepsin D, (**B**) cathepsin L and (**C**) cathepsin B during marinating. [a,b,c] Results marked with the same lower case letter do not statistically differ by the influence of marinating time; [A,B,C] Results marked with the same capital letter do not statistically differ by the influence of the marinating method. For a better presentation of the results, a break ($\approx$) in the Y axis was used.

Cathepsin L activity, after 2 days of marinating the fillets, increased twice to 612–683 U, after which it decreased to 170–380 U (Figure 6B). In marinades from headed herring, cathepsin L activity increased 6.5 times after 2 days by the German method (1679 U) or after 7 days by the Danish method (1692 U). In marinades from whole herring, cathepsin L activity increased as much as 8 and 9 times after 4 days in the Danish (2000 U) and German (2260 U) methods, respectively, and then decreased, reaching 1400–1499 U after 14 days of marinating. Cathepsin L, like cathepsin D, is also an acidic endopeptidase, but requires the presence of salt and a higher pH for optimal activity [2]. Therefore, cathepsin L activity positively correlated with the pH value in only G-fillets (0.593), G-headed (0.964) and G-whole (0.736) marinades. In turn, salt concentration significantly correlated with cathepsin L activity for all marinades, except G-fillets.

In the case of cathepsin B, the activity of this endopeptidase decreased 20–40 times after 2–7 days of marinating; the fastest in G-fillets marinades and the slowest in D-whole marinades (Figure 5C). The cause was the strong inactivation of cathepsin B by salt and acetic acid [2]. Thus, the results showed that the activity of digestive proteases and cathepsins was higher in marinades made from whole and headed herring than from fillets, especially after 2–7 days of marinating. The higher cathepsin activity in marinades from ungutted herring was possibly related to (i) the release of cathepsins from lysosomes by digestive proteases [6,34], (ii) the reduction of cathepsin loss from meat to brine by the

herring skin [2] and (iii) the slower diffusion of acetic acid into meat [32], which inhibits the activity of alkaline digestive proteases. Moreover, higher cathepsin D activity may have promoted the release of intracellular proteases [35].

### 3.3. Concentration of Protein Hydrolysis Products (PHP)

The process of enzymatic ripening of herring meat leads to the formation of TCA-soluble proteins, peptides and amino acids, the quantitative composition of which allows assessment of the dynamics of the ripening process of marinades [3,36]. The average NPN content of fillets marinated for 2–14 days was significantly lower by 66–91% than in marinades from whole or headed herring (Figure 7A). NPN content in fillets after 2 days decreased to 156 and 117 mg in the German and Danish methods, respectively. The reason for the decrease in NPN content in the German method was the diffusion of NPN from the meat into the marinating brine [30], while in the Danish method, there was an additional increase in moisture during the salting stage. As a result, fillets marinated for 2 days contained 16–18% less NPN than the raw material. This phenomenon did not occur in whole or headed herring, where diffusion was limited by the skin, causing these marinades to contain 344–356 mg of NPN after 2 weeks, 60–65% more than the raw material (Figure 7A). The higher NPN content was also due to higher proteases activity in marinades form whole and headed herring.

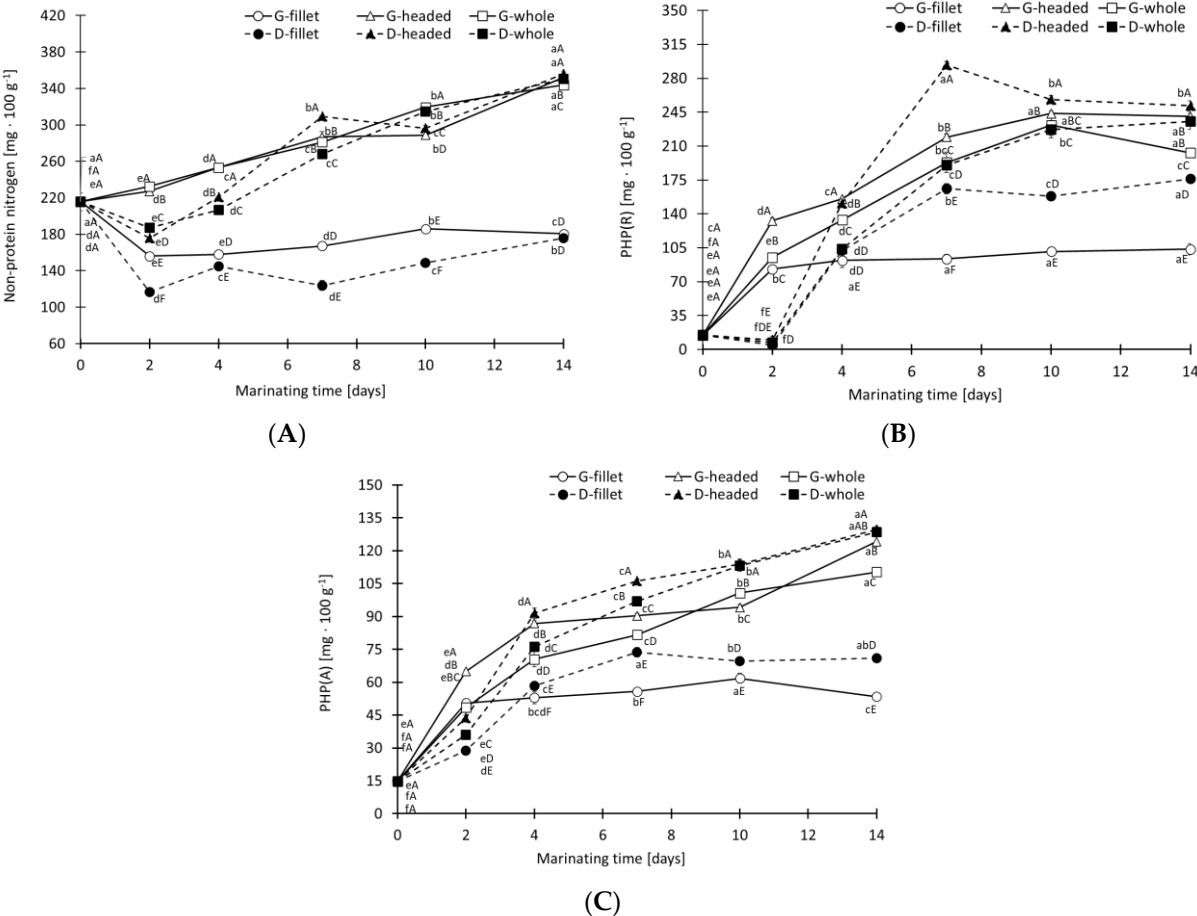

**Figure 7.** Effect of marinating method (G—-German, D—-Denmark) and degree of primary processing of Baltic herring (whole, headed, fillet) on concentration of (**A**) NPN, (**B**) PHP(R) and (**C**) PHP(A) during marinating. [a,b,c] Results marked with the same lower case letter do not statistically differ by the influence of marinating time; [A,B,C] Results marked with the same capital letter do not statistically differ by the influence of the marinating method.

The peptide fraction content of G-fillets marinades significantly increased only up to the fourth day, reaching 92–103 mg (Figure 7B), which is characteristic for the German method [32]. On the other hand, during marinating of whole and headed herring, the peptide content increased up to the tenth day, reaching 232–234 mg PHP(R). In the Danish method, the peptide content decreased by a few mg after 2 days due to an increase in the moisture of the salted meat, after which it sharply increased to 7 days in fillets and headed herring or to 14 days in whole herring. After 7–10 days, marinades from fillets and headed herring obtained by the Danish method contained more peptides compared to the German method (Figure 7B). Peptide content was not significantly different after 7–10 days for whole herring marinated using both methods. This shows that the inhibition of cathepsin D by salt, and the lower dynamics of mainly peptide formation in the Danish method than in the German method, were balanced by high digestive protease activity. Additionally, the content of the amino acid fraction changed during marinating, similar to the content of NPN and peptides. After 7 days of marinating, G-fillets marinades contained the least PHP(A) (55.8 mg), followed by D-fillets (73.7 mg), G-whole (81.6 mg), G-headed (90.3 mg), D-whole (96.9 mg), and the most, D-headed (106.1 mg) (Figure 7C). For the amino acid fraction, the increase in content was characteristic up to 7–10 days in fillets, while up to 14 days in marinades from ungutted herrings.

Statistical analysis showed that NPN, peptides and amino acid content mainly correlated in G-fillets marinades with cathepsin D (0.637–0.931) and Am-Tp (0.633) activity, in G-headed marinades with the addition of chymotrypsin activity (0.610–0.705) and in G-whole marinades with the addition of Es-Tp and Cp-A activity (0.580–0.946) (Figure 3A,C,E). For the Danish method, PHP content mainly correlated in D-fillets marinades with the addition of cathepsin D activity (0.565–0.746), in D-headed marinades with the addition of cathepsin L (0.704–0.935), and in D-whole marinades with Es-Tp (0.725–0.815) (Figure 3B,D,F). Thus, the lower degree of preliminary processing of herring promoted more significant correlations of digestive proteases with PHP and with higher cathepsin D activity. The results showed a synergistic effect of digestive and muscle enzymes in the proteolysis of marinated herring proteins (Figure 3), confirming the results of Kamiński et al. [7].

### 3.4. Hardness-TPA and Sensory Assessment of Marinated Meat

For two weeks of marinating, the highest texture rating in sensory analysis was regularly attributed to marinades made using the German method; with the exception of the 14th day in the case of marinades made from headed herring (Figure 8A). G-whole marinades scored highest in texture, with scores ranging from 4.79 to 4.93 after 4–14 days of marinating. D-fillets received the lowest marks of 4.20–4.43 for texture. German marinades needed only 4–7 days to achieve maximum marks for texture, while Danish marinades needed at least twice as long. Texture profile analysis (TPA) showed that the hardness-TPA of the meat of G-whole and G-headed marinades averaged 6.5 N, while the hardness of Gs marinades was significantly higher, at 9.6 N (Figure 8B). On the other hand, in the case of the Danish method, the hardness-TPA of the D-headed and D-whole marinades increased to from the fourth and decreased by the tenth day, obtaining higher values than the German G-headed and G-whole marinades (except D-headed after the seventh day). This could have been due to the lower activity of cathepsins L and D in the Danish marinades than German marinades. Cathepsin L hydrolyzes collagen, which has a greater impact on meat hardness than muscle proteins [37], which are mainly hydrolyzed by cathepsin D [38]. It was observed that cathepsin L activity positively correlated with texture in marinades obtained from herring with intestines (G-headed 0.847, D-headed 0.861, D-whole 0.779), contrary to a negative correlation in marinades from fillets (G-fillets −0.709, D-fillets −0.772). Although both marinating methods had a positive effect of marinating with digestive proteases on improving meat texture, significant hardness-TPA correlations with texture in sensory analysis were only for G-whole (0.620) and D-whole (0.568) marinades (Figure 3E,F).

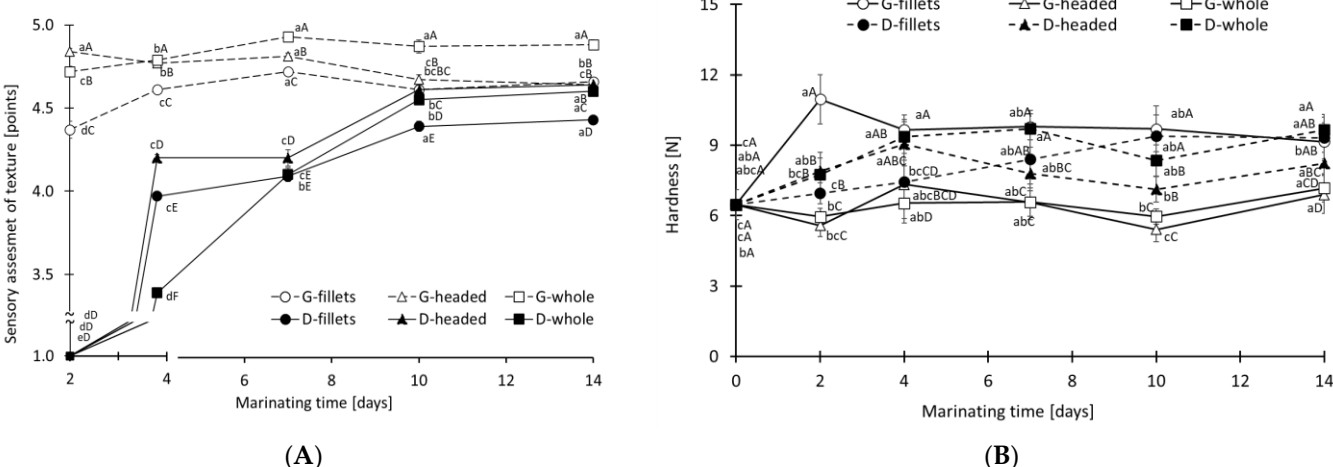

**Figure 8.** Effect of marinating method (G—-German, D—-Denmark) and degree of primary processing of Baltic herring (whole, headed, fillets) on (**A**) texture sensory evaluated and (**B**) hardness evaluated by TPA. [a,b,c] Results marked with the same lower case letter do not statistically differ by the influence of marinating time; [A,B,C] Results marked with the same capital letter do not statistically differ by the influence of the marinating method. For a better presentation of the results, a break ($\approx$) in the Y axis was used.

Additionally, the results of sensory analysis showed that a bitter taste appeared in the D-whole and D-headed marinades after 4 days of ripening, the intensity of which increased up to the tenth day of marinating from 1.0 to 4.0 points and from 0.5 to 2.0 points, respectively. The bitter taste was not present in D-fillets marinades, indicating that it was caused by the intestines of the fish. In the case of the German method, the bitter taste also appeared, but only after 10–14 days of marinating in G-whole and G-headed marinades, obtaining 1.0 and 0.5 points, respectively. The bitter taste could have come from blood and/or from the gastrointestinal tract and/or from bitter peptides, which are mainly released by chymotrypsin [39]. Chymotrypsin activity after 10–14 days of marinating was highest in only the D-whole and D-headed marinades, then was significantly lower in G-whole and G-headed marinades, and was lowest in fillets marinades (Figure 5C). In turn, carboxypeptidase-A, which is known to reduce bitter taste [40], was most active in G-whole marinades (Figure 5D), which likely protected these marinades from the rapid appearance of bitter taste.

### 3.5. Microbiota of Herring

Contamination of fresh herring meat with psychrophilic bacteria was 6.60 log cfu/g, mesophilic bacteria < 0.1 log cfu/g, moulds and yeasts < 0.1 log cfu/g and bacteria Enterobacteriaceae 3.00 log cfu/g (data not shown). After marinating, microbiological contamination of the four groups of bacteria was below 0.1 log cfu/g, except for psychrophile contamination after 2 days in Danish marinades: D-fillets 5.46, D-headed 6.30 i D-whole 5.28 log cfu/g. Microbiological analysis showed that the addition of acetic acid completely inhibited the proliferation of microorganisms in marinated meat, including Enterobacteriaceae from the digestive tract of the herring. Acetic acid, among other organic acids used in marinades, has the strongest bacteriostatic effect [41]. Kamiński et al. [7] showed that Enterobacteriaceae bacteria were the most resistant to marinating, but did not pose a threat to the microbiological quality of the marinades.

### 4. Conclusions

Studies have shown that marinating whole herring using the German method increased the mass yield of marinated fillets by several percent (10–15% after 4–14 days of marinating), which was also possible using the Danish method. However, the Danish

method increased the salt content by 0.7–1.7 percent after 2 days of marinating, while in congtrast, the German method inhibited cathepsin D by 32–55% and promoted a bitter taste. The technology of marinating with viscera made it possible to reduce the ripening time of marinades by 3 days, from 7–10 days to 4–7 days. Diffusion of viscera proteases into the meat increased viscera proteases activity and cathepsin activity, which made it possible to obtain well-ripened meat as early as 4–7 days. Fillets from the same herring had mainly cathepsin activity, and did not achieve full and proper meat ripeness even after 14 days. In addition, marinating of ungutted herring promoted a longer and greater increase in the content of protein hydrolysis products (PHP), which give marinades their characteristic flavor. Therefore, especially important was the twice higher carboxypeptidase A activity in G-whole marinades (8.3–9.5 U) compared to G-headed and G-fillets (4.3–5 U). The 2–3 times higher PHP content for whole and headed samples, compared to fillets samples, was due to two phenomena: higher proteolytic activity in the meat and reduced PHP loss from the meat to the brine.

The results showed that the contribution of cathepsin D to meat ripening can be supplemented or even largely replaced by digestive proteases, but the right conditions for their activity (faster or previous diffusion of proteases than acetic acid and/or faster or previous diffusion of salt than acetic acid) must be applied. This is crucial knowledge in the field of marinating herring of low technological quality or non-herring fish resistant to marinating, whose meat does not reach full ripeness.

**Author Contributions:** Conceptualization, methodology, formal analysis, investigation writing—review and editing P.K., M.S. and B.S.; visualization P.K.; All authors have read and agreed to the published version of the manuscript.

**Funding:** This work was supported by Rector of the West Pomeranian University of Technology in Szczecin for PhD students of the Doctoral School, grant number: 540/080—Patryk Kamiński.

**Institutional Review Board Statement:** Not applicable.

**Informed Consent Statement:** Not applicable.

**Data Availability Statement:** Not applicable.

**Conflicts of Interest:** The authors declare no conflict of interest.

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
