# Peer review of "Effects of Primary Processing and Pre-Salting of Baltic Herring on Contribution of Digestive Proteases in Marinating Process"

_applsci, doi:10.3390/app122110877_

Round 1
Reviewer 1 Report
To Authors,
in their paper, titled "Effects of primary processing and pre-salting of Baltic herring on contribution of digestive proteases in marinating process", authors Patryk Kamiński et al. studied the German and the Danish marination processes applied to whole, headed and fillleted Baltic herring. They found that " Marinating with intestines [whole fish] makes it possible to produce marinades faster, more efficiently and with higher sensory quality from herring of low-technological quality."
The manuscript reads well. It reports an interesting study, that continues authors' previous research [refs. 1-6], addressing a "traditional" method of transformation of fish and contributing to its optimization/modernization. The findings have real-life applications that might increase readership. However, a few instances require clarification and/or revisions.
Minor issues:
-
L. 53, 82, 91, 129, 175, 181, 194, 197, 198, 213, 336
Completion/Completion:
-
L. 101 Which AOAC techniques? Complete.
-
L. 130-131 Units? N.
-
L. 134 How many panelists? Add few more details about training and/or expertise of the panelists.
-
L. 135 Which were? More details about the actual labels is needed.
-
L. 139 or of the marinated fillets? Clarify.
-
L. 144-146 Add a few more details about these (microbiol.) methods.
-
L. 149 ANOVA level of significance instead of P-value => in "classical" hypothesis testing, the level of significance, alpha, is set a priori and the p-value - a probability derived/"associated" with the test statistic (in this case F - is usually compared with alpha to make a decision about the hypotheses)
-
L. 156 This section should be renamed Results and Discussion since authors end up discussing their findings even though sparingly (L. 159, 198-201, 231-234, 247-248, 252-256, 262-264, 272-277, 286-287,291-294,...)
-
In Fig 3. Why parameters not measured on day 0? Make ref to Y-axis break in the caption.
-
L. 213 Relatively clear fo G-whole but not for G-headed of G-fillets which was in fact the lower on day 14 in Fig 4? Clarify.
-
FIg. 4 Make ref to Y-axis break in the caption.
-
Possibly the (very) long paragraph (L. 242-277) could be shortened i.e. all the details described in per-group approach could be summarized.
-
Fig. 6 (on pages 8 and 9) is actually Fig. 5? Why no data for Day 0? In its caption check the labels A-C. Make ref to Y-axis break in the caption.
-
Missing Fig. 6 about PHP.
-
Fig. 7 What is the correposndence of these values (in A) with scale of sensory assessment? See comment to Subsec. 2.6. in A, Why no data for Day 0? Make ref to Y-axis break in the caption.
-
L. 345 Not that clear the pattern in Fig.7
-
L. 358, Why (data not shown)? this could be relevant for products acceptance discussed next?
-
L. 368, Microbiota instead of microflora.
-
L. 370 use always "complete" units log cfu/g.
-
L. 385, "several percent" Complete/Clarify.
-
L. 380 In this section (4. Conclusions) I'd expect statement of your conclusions - of this work - with actual (more relevant) values instead of more general "fiinal considerations". Consider revising.
These are included in the PDF file using the commenting tools of Adobe Reader DC.

Reviewer 2 Report
The present manuscript deals with the novel area of improving the quality of herring meat by using its gut/ proteases. The manuscript is well-written and presented some interesting findings that may have future applications in the fish industry. The topic is very relevant to the present focus on by-product utilization. The language is clear and easy. The hypothesis is well-stated and clear.
My observations are as follows-
I think a suitable diagrammatic/ figure/ tabular representation of Germany vs Danish methods or the experimental design of this experiment will improve the citation and readership of this article.
1. Abstract: Appropriate
2. Keywords: Appropriate
3. Introduction: well-presented sufficient information and justification for the start of the work. L39-40: I can understand from the sentence that herring in a heavy feeding state had the lowest cathepsin activity? If yes then please make it clear.
4. L 44- show may be slow
5. L 82: the scientific name in italics plz
6. L 85-87- plz cite references if data from other studies/ sources
7. L 154: please mention n=?
8. Figures: excellent and informative
9. Results and Discussion: Appropriate and well supported by relevant literatures.
